# Electric Source Imaging in Presurgical Evaluation of Epilepsy: An Inter-Analyser Agreement Study

**DOI:** 10.3390/diagnostics12102303

**Published:** 2022-09-24

**Authors:** Pietro Mattioli, Evy Cleeren, Levente Hadady, Alberto Cossu, Thomas Cloppenborg, Dario Arnaldi, Sándor Beniczky

**Affiliations:** 1Department of Neuroscience (DINOGMI), University of Genoa, 16132 Genoa, Italy; 2Danish Epilepsy Center, 4293 Dianalund, Denmark; 3Department of Neurology, University Hospital Leuven, 3000 Leuven, Belgium; 4Department of Neurology, Albert Szent-Györgyi Medical School, University of Szeged, 6720 Szeged, Hungary; 5Child Neuropsychiatry, Department of Surgical Sciences, Dentistry, Gynecology and Pediatrics, University of Verona, 37126 Verona, Italy; 6Department of Epileptology, Krankenhaus Mara, Medical School, Bielefeld University, 33615 Bielefeld, Germany; 7IRCCS San Martino Hospital, 16132 Genoa, Italy; 8Department of Clinical Neurophysiology, Aarhus University Hospital, 8200 Aarhus, Denmark; 9Department of Clinical Medicine, Aarhus University, 8000 Aarhus, Denmark

**Keywords:** EEG, epilepsy, presurgical evaluation, source analysis, source imaging

## Abstract

Electric source imaging (ESI) estimates the cortical generator of the electroencephalography (EEG) signals recorded with scalp electrodes. ESI has gained increasing interest for the presurgical evaluation of patients with drug-resistant focal epilepsy. In spite of a standardised analysis pipeline, several aspects tailored to the individual patient involve subjective decisions of the expert performing the analysis, such as the selection of the analysed signals (interictal epileptiform discharges and seizures, identification of the onset epoch and time-point of the analysis). Our goal was to investigate the inter-analyser agreement of ESI in presurgical evaluations of epilepsy, using the same software and analysis pipeline. Six experts, of whom five had no previous experience in ESI, independently performed interictal and ictal ESI of 25 consecutive patients (17 temporal, 8 extratemporal) who underwent presurgical evaluation. The overall agreement among experts for the ESI methods was substantial (AC1 = 0.65; 95% CI: 0.59–0.71), and there was no significant difference between the methods. Our results suggest that using a standardised analysis pipeline, newly trained experts reach similar ESI solutions, calling for more standardisation in this emerging clinical application in neuroimaging.

## 1. Introduction

Refractory epilepsy has been defined by the International League Against Epilepsy as the persistence of seizures even after adequate trials with two or more appropriate and tolerated anti-seizure drugs [1]. The occurrence of seizures lowers the quality and quantity of life in affected patients and increases the risk of sudden unexpected death by epilepsy [2]. For patients with refractory focal epilepsy, resective surgery is an evidence-based treatment option to obtain seizure freedom [3,4]. Epilepsy surgery is only successful if the so-called epileptogenic zone can be reliably identified and safely resected. In order to estimate the location and extent of the epileptogenic zone, patients undergo a multimodal presurgical workup, including long-term video-EEG monitoring, structural (MRI) and functional (FDG-PET, ictal/interictal SPECT, fMRI, DTI) neuroimaging, neuropsychological testing and occasionally invasive EEG monitoring [3,5]. A relatively new and underused technique during the presurgical evaluation is electric source imaging (ESI).

ESI is a neuroimaging technique that allows one to estimate the location in the brain of the source generating the electroencephalography (EEG) signals registered by scalp electrodes [6]. To perform ESI, the forward and the inverse problems have to be considered [7]. By solving the former, a leadfield is calculated by combining the exact positions of EEG electrodes with structural information of the patient’s own MRI [7]. The latter is to identify the location of an unknown source by analysing a scalp EEG signal and has been defined as an ill-posed problem [6]. In fact, several combinations of sources can theoretically generate the same EEG signal [8], and therefore, it is necessary to make assumptions [6,7,9]. Several inverse solutions have been developed during recent years, and they can be classified into two groups based on the presence (equivalent current dipole (ECD)) or the absence of the assumption that the source is generated by one or a few brain areas (distributed source model (DSM)) [9].

Despite the complexity of the theoretical background, ESI was proven to be particularly useful in the presurgical evaluation of patients with drug-resistant focal epilepsy, because it provides non-redundant information that can change a patient’s management and outcome [10,11,12]. For instance, it provides additional data about the possible epileptogenic zone location, helps to plan stereo-EEG implantations and allows doctors to proceed with the presurgical evaluation of those patients who would otherwise have been identified as ineligible [7,10,11,13].

The inverse solution method, the number of electrodes (low density, extended scalp array, high density or very high density), the signal (ictal or interictal) and the software used to perform ESI are all factors that have to be considered when performing ESI [7,13,14,15,16,17,18]. Therefore, several studies were conducted to explore differences between various inverse solutions [14,15], interictal and ictal source imaging [13], low and high density EEG [16,17,18] and different software [10]. However, in clinical practice, in spite of a standardized analysis protocol, ESI includes steps with subjective decisions about the individual patient by the analyser, i.e., the expert doing the ESI. These subjective decisions are related to the selection of the interictal epileptiform discharges (IEDs) to be analysed, identification of the IED-onset epoch and seizure onset identification, particularly if a completely automated ESI is not used [15,19].

Obviously, this method would be of little or no clinical use if different experts reached different results when analysing the same recordings, using the same pipeline and software. Nonetheless, this important aspect has not been investigated yet. The aim of this study was therefore to investigate the degree of inter-analyser agreement of ESI performed using a standardized pipeline and software certified for medical use, by young experts without previous experience in performing ESI.

## 2. Materials and Methods

### 2.1. Patients and EEG Recordings

Twenty-five patients (median age 32 years, range 12–63; 13 females) who underwent presurgical evaluation for drug-resistant focal epilepsy were consecutively enrolled at the Danish Epilepsy Centre (Dianalund, Denmark). There were 17 patients with temporal lobe epilepsy and 8 with extratemporal lobe epilepsy. The study protocol was approved by the regional Ethics Committee (SJ-722; 24 September 2018), and participants signed an informed consent form in compliance with the Helsinki Declaration of 1975.

All patients underwent long term video-EEG (LTM) recording in the epilepsy monitoring unit, where at least one seizure was detected. LTM EEG was recorded using 40 scalp electrodes, in accordance with the 10-10 setting, by adding to the standard IFCN 25-channel electrodes [20] eight intermediate electrodes (FC1, FC2, FC5, FC6, CP1, CP2, CP5, CP6) and seven inferior electrodes (AF1, AF2, PO1, PO2, IO1, IO2, Iz) [16]. Subsequently, high density EEG (HD) was performed in 15 patients (median age: 29 years, range 12–61; 7 females). HD EEG was recorded using 256 scalp electrodes evenly distributed on the head and neck. Ten patients did not undergo HD registration because of the low number of IEDs (<1/h) during the LTM. In the case of absence of recorded IEDs during the LTM, which was the case for seven patients, LTM analysis was not performed. Seizures were recorded only during the LTM EEG recordings.

### 2.2. Electric Source Imaging Pipeline

Anonymized EEG recordings were retrospectively analysed, blinded to all clinical data, using BESA Research software v7.1 (BESA, Gräfelfing, Germany). Six experts (S.B., T.C., A.C., L.H., E.C., P.M.) independently applied the same analysis pipeline. Selection of IEDs, of ictal waves and of the signal time window were analyser-dependent, as these aspects are tailored to the individual patient. S.B. had previous experience (18 years) in ESI, and the other analysers were introduced to the methodology at the start of this study. Two of them (T.C., E.C.) were senior experts with more than five years of experience in EEG reading. Three analysers were junior experts with less than five years of experience in EEG reading (A.C., P.M., L.H.). All experts had their EEG training at various centres in Europe. MRI segmentation and individual head models using finite element method (FEM) leadfield were generated with BESA MRI 3.0 (BESA, Gräfelfing, Germany). Interictal and ictal pipelines are illustrated in Figure 1 and Figure 2, respectively. We start the description of the pipeline with the generation of the head model, as it will be used in the subsequent paragraphs to show the calculation of the inverse solutions.

#### 2.2.1. FEM Leadfield Generation

As part of the presurgical evaluation, brain MRI was acquired for all patients following international guidelines [21]. Three-dimensional T1-weighted isotropic scans were used for segmentation. 

Semi-automated MRI segmentation was performed on each brain MRI after manual selection of 14 cardinal points: anterior commissure; posterior commissure; anterior, posterior, superior, rightmost and leftmost marginal planes; nasion; inion; and five brainstem markers (Figure 1C and Figure 2C). After segmentation, the individual FEM head model was calculated and co-registered with digitalized [22] electrode positions to obtain the FEM leadfield (Figure 1D and Figure 2D).

#### 2.2.2. Interictal ESI

First, after a quality check of the signal, IEDs were visually identified (Figure 1A). IEDs with the same voltage distribution at the peak were defined as belonging to the same cluster. For each cluster, at least five IEDs were manually annotated and averaged to serve as a template. Then, an automated pattern search was performed over all channels using a 2–35 Hz filter and a threshold of 85% similarity. The results of the automated pattern search were visually evaluated and edited when needed. The accepted IEDs of each cluster were averaged to increase the signal-to-noise ratio (Figure 1A). 

Next, sequential voltage maps were constructed on the ascending slope of the averaged IED to assess intraspike propagation (Figure 1B). If propagation was detected, the time point at which the first stable distribution was seen (i.e., the onset of the IED) was chosen for further analysis. In the BESA Research source analysis module, principal component analysis (PCA) was performed on the signal to further evaluate intraspike propagation and to select the length of the time window for the inverse solution (Figure 1B). The analysis interval ranged from the time point chosen from the sequential voltage maps to a time point which would give rise to a PCA of more than 95%, implying that at least 95% of the signal can be explained by one source, as 5% can be caused by noise, even after averaging. If no intraspike propagation was detected, the analysis time interval was chosen as the middle third of the ascending slope [12] by taking into account the PCA. 

Finally, by combining the analysis time interval with the previously calculated FEM leadfields, the inverse solutions were calculated. Two different inverse solutions were applied using the available functions in BESA Research (Figure 1E): ECD and a DSM restricted to the cortex (cortical CLARA) [14]. ECD was performed in the interval as described above. As the DSM is more sensitive to noise, the cortical CLARA was calculated at the peak if there was no clear propagation. If clear propagation was detected, the analyser had to choose the time window that allowed him to fit the onset of the cluster with the higher signal-to-noise ratio.

#### 2.2.3. Ictal ESI

The electrographic seizure onset was determined visually by each expert, for the same seizures. First, each ictal wave with a similar voltage distribution of the first distinguishable ictal rhythmic activity was marked on its negative peak (Figure 2A). Ictal waves from which the voltage distribution was distorted by artefacts were discarded. Ictal waves with the same voltage distribution but different frequencies were considered similar. Then, manually selected waves were averaged to increase the signal-to-noise ratio. 

Next, as for the interictal analysis, sequential voltage maps were evaluated (Figure 2B). Due to the higher level of noise in the ictal averaged signal compared to the interictal signal, especially when averaging similar waveforms at different frequencies, the onset of the ictal wave would generally contain too much noise to be modelled properly. Therefore, both ECD and DSM analysis were performed at the peak of the averaged wave (Figure 2E). 

### 2.3. Sublobar Localisation

ESI results from each analyser were reviewed by two experts (E.C. and P.M.) to localize the sources at a sublobar level. In case of disagreement, a third expert (S.B.) intervened for the final localisation.

The localisation of ESI results consisted of 50 classifiers, including 46 sublobar regions [23] (Appendix A). If the source location was not clear, multifocal (without a clear dominant source) or outside the cerebrum, three additional classifiers (non-localizable and left or right non localizable) were available. To reproduce real-world clinical practice, when a source was located on the border of a clear lesion or an old resection cavity, the classification “perilesional” was used, regardless of sublobar classification.

### 2.4. Statistical Analysis

To summarize, each analyser performed ictal ESI on every patient and interictal ESI (both LTM and HD) when sufficient IEDs were present in the recordings. 

Analysers were free to choose the IEDs to analyse, and inter-analyser agreement was calculated between IEDs with the same temporal and spatial topography. Moreover, due to the heterogeneity of focal epilepsy, sources were not evenly distributed between the different sublobar regions. For these reasons, inter-analyser reliability was calculated by applying Gwet-AC1 [24] to avoid the Kappa paradox [25]. 

The inter-analyser agreement was considered as poor (agreement coefficient, AC < 0), slight (0.01 < AC < 0.2), fair (0.2 < AC < 0.4), moderate (0.4 < AC < 0.6), substantial (0.6 < AC < 0.8) or almost perfect (AC > 0.8), as described in previous studies [10,26].

Statistical analyses were performed in R 4.1.3. using packages irrCAC (v1.0) and ggplot2 (v3.3.5).

## 3. Results

Figure 3 illustrates the ESI results for a patient with temporal lobe epilepsy (Figure 3A) and a patient with extratemporal lobe epilepsy (Figure 3B). 

Overall interrater agreement was substantial (AC1 = 0.65; 95%CI: 0.59–0.71) and remained substantial when evaluating ECD and DSM separately (Table 1 and Figure 4). There was no statistically significant difference between the modalities. However, interrater agreement tended to be highest for the interictal LTM ESI, with substantial agreement, and lower for ictal analysis, with moderate agreement overall, but the latter had substantial agreement with the ictal ECD. Table 1 and Figure 4 show an overview of the interrater agreements between all the combinations of inverse solutions and recordings.

## 4. Discussion

In this study, we evaluated the inter-analyser reliability of ESI in a group of 25 patients with refractory epilepsy who underwent presurgical evaluation. Our results show that when applying the exact same analysis method, despite some inevitable subjective decisions, the inter-analyser agreement is substantial (AC1 = 0.65; 95%CI: 0.59–0.71). These results are of utmost importance for the application of the ESI in clinical practice, as this proves that this method is robust, reliable and reproducible. 

Several studies have been conducted to provide clinicians information about ESI methodology and to assess its reproducibility in clinical practice. Nonetheless, previous studies did not focus specifically on the reliability of the method between analysers and did not use a specific pipeline to evaluate the residual reliability produced by the subjective decisions of the individual analysers. Duez and colleagues evaluated the clinical utility of ESI during the presurgical workup for epilepsy [10]. To evaluate the accuracy in localising the source of IEDs, they compared the results of ESI and MSI performed by two analysers. However, although each of them analysed the same HD-EEG signals using ECD and a DSM, each used a different software package and DSM. Even if they found substantial agreement between the two inverse solutions within each software package, the agreement between the different software and the different analysers, in both ECD and DSM, was only moderate. Therefore, the authors suggested that further standardisation was needed to increase the agreement in source localisation [10]. 

Considering literature data, it appears that the variability in source localisation is not related to the decision on which inverse solution is used. In fact, our group previously showed good agreement at the sublobar level between five different inverse solutions in ictal HD-EEG signals [14]. Accordingly, in this study we did not find significant differences in the interrater agreement coefficient between ECD and DSM in ictal ESI and interictal HD and LTM ESI. We obtained lower, not significant, agreement for the ictal DSM, most likely because of the lower signal-to-noise ratio of the ictal signal.

Interestingly, in our sample, no differences between HD-ESI (256 electrodes) and LTM-ESI (40 electrodes) were found. This is not surprising, considering that LTM recordings are far longer than HD recordings. Thus, the higher number of IEDs detected counterbalanced the lower number of channels, providing a similar amount of data for principal component analysis and dipole fitting. In fact, to identify a higher number of IEDs, a longer recording has been proven to be more important than a larger number of electrodes [27]. Nevertheless, it should be considered that in this study, 40 channels were used for the LTM, thereby significantly increasing the number of signals compared to the standard 25 electrodes. 

The aforementioned studies provided important information about ESI methodology reliability but did not focus on the subjective decisions that the analyser has to take during ESI. In this study, we aimed to mimic the clinical practice by having the analyser perform ESI while blinded to other clinical information. Thus, the analysers used a standardised pipeline that included two inverse solutions and the use of the same software package (in this case, BESA research 7.1), but they were free to choose the IEDs to analyse, the seizure onset and the time window for fitting the dipoles. This way, the inter-analyser residual disagreement was to be the consequence of the individual choice of the expert and not of the method. Even with these subjective decisions, we obtained substantial inter-analyser agreement independently of the inverse solution method, the analysed signal and the number of EEG channels. Therefore, a limited number of subjective decisions does not affect the reliability of the method when following a standardised pipeline. Notice, even if not statistically significant, the ictal agreement coefficient tended to be lower than the interictal agreement coefficient. This was an expected result, because the decision of the seizure-onset window relies on the individual experience of the analyser. Nevertheless, when evaluating the ECD for the ictal ESI, the agreement coefficient remained substantial, proving that reliability is high for ictal ESI too, if ECD is used as an inverse solution method.

This study has some limitations. First, the sample size was relatively small; therefore, the results should be confirmed by larger, multicentric studies. Second, five of the experts had their ESI training at the same centre. This was, however, part of the aim of our study, as we aimed at verifying the reliability of the methodology when applying the same pipeline while allowing subjective decisions that relied on the levels of EEG experience of the experts (which was gained at different European centres). Third, the same software was used to perform the analysis, but the reliability of the use of different software in performing ESI has already been explored in a previous study by our group [10].

Routine visual interpretation of EEG provides “only” moderate agreement between analysers, according to previous studies [28,29]. Therefore, to provide high quality diagnosis and treatment and for research purposes, standardisation is of crucial importance in clinical neurophysiology. Many efforts have been made to standardise EEG terminology [30], reading [31,32] and reporting [23]. Nevertheless, in clinical practice and in particular in ESI, a limited number of subjective decisions remain. Therefore, it is important to understand how much those decisions influence the reliability of a method. Here, we make a significant step towards this goal in ESI, by proving that by using a standardised pipeline in the application of ESI, the limited residual subjective decisions do not significantly alter the agreement between the ESI analysers.

## Figures and Tables

**Figure 1 diagnostics-12-02303-f001:**
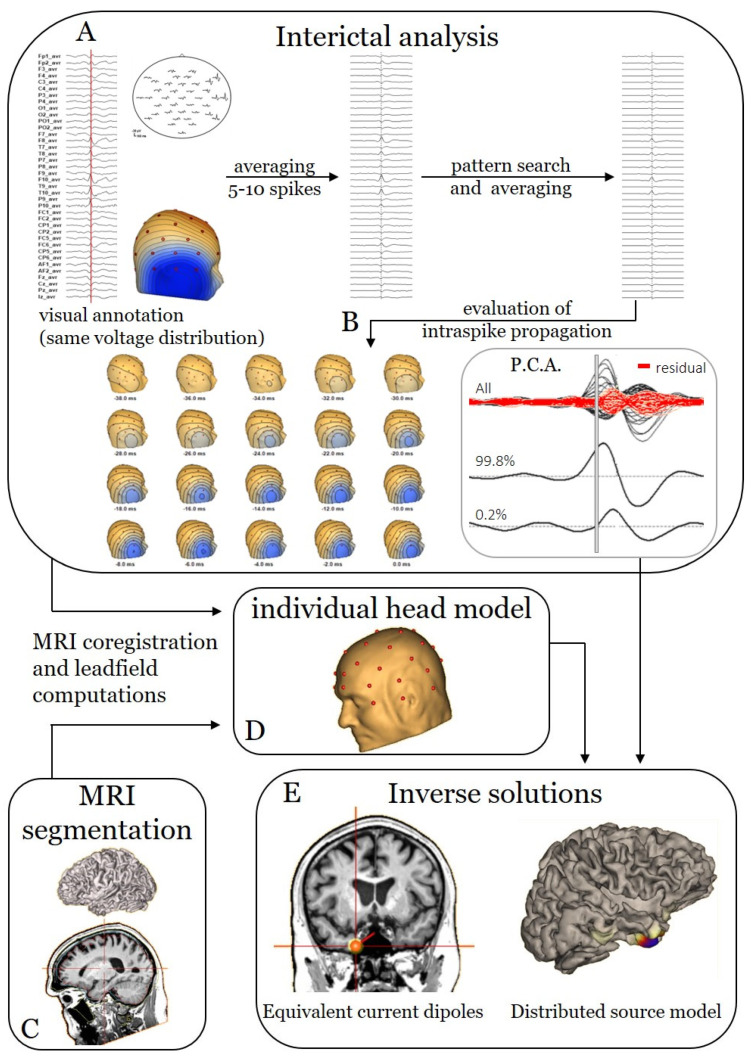
Analysis pipeline for interictal ESI. (**A**) First, a minimum of five IEDs of the same cluster (i.e., having the same voltage distribution) are visually annotated. Averaging of these IEDs creates a template which is used to perform a pattern search throughout the EEG file. The automatically detected IEDs are averaged to result in a better signal-to-noise ratio. (**B**) Next, intraspike propagation is assessed by visual analysis of the sequential voltage maps and a principal component analysis of the signal. (**C**) The patient’s MRI is segmented into different tissue classes; the electrodes are aligned to the scalp and are used to solve the forward problem, i.e., the creation of an individual head model (**D**). (**E**) Finally, source modelling is performed using two different inverse solution methods: equivalent current dipoles (ECD); a distributed source model (DSM), cortical CLARA.

**Figure 2 diagnostics-12-02303-f002:**
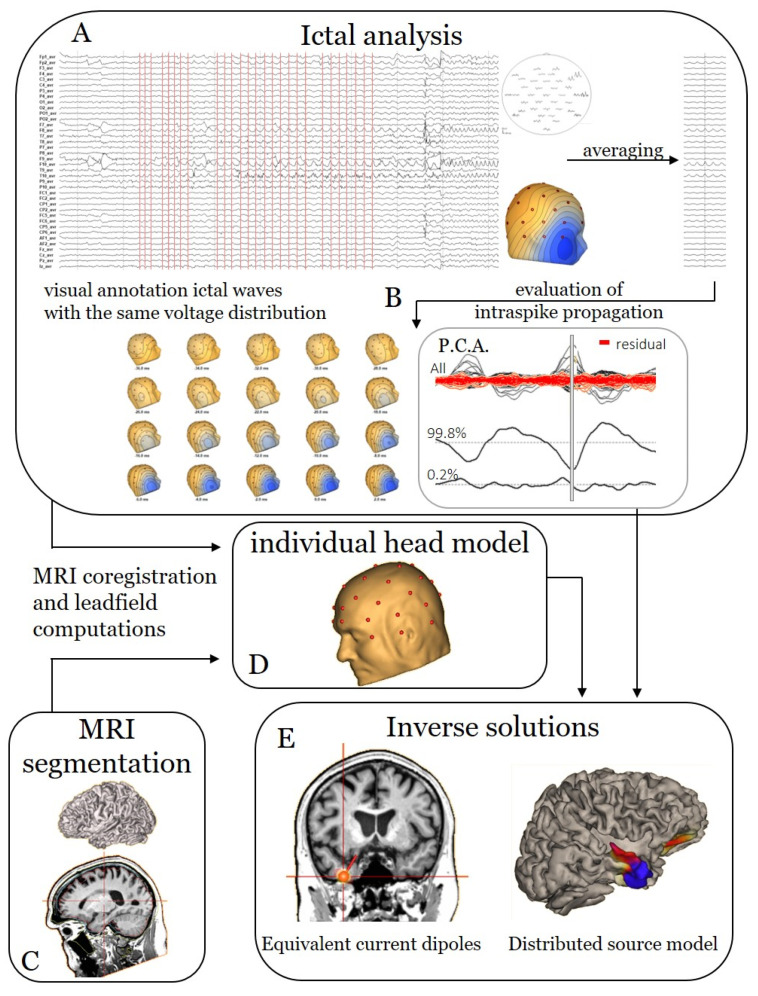
Analysis pipeline for ictal ESI. (**A**) The electrographic onset of the seizure is identified and marked manually. The first ictal waves with the same voltage distribution but without artefacts are annotated at their peaks. Next, these waves are averaged to increase the signal-to-noise ratio. (**B**) As these signals contain more noise than the interictal averaged signal, the signal is analysed at the peak. (**C**) The patient’s MRI is segmented into different tissue classes; the electrodes are aligned to the scalp and are used to solve the forward problem, i.e., the creation of an individual head model (**D**). (**E**) Finally, source modelling is performed using two different inverse solution methods: equivalent current dipoles (ECD); a distributed source model (DSM), cortical CLARA.

**Figure 3 diagnostics-12-02303-f003:**
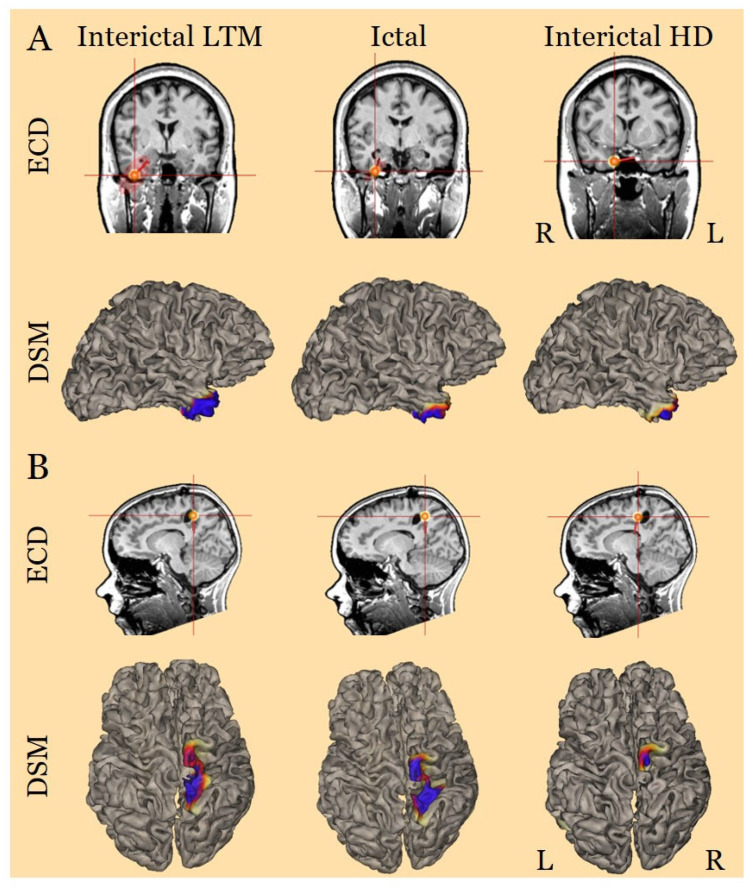
Examples of temporal (**A**) and extratemporal (**B**) sources. (**A**) Interictal LTM, ictal and interictal HD sources from a patient with temporal lobe epilepsy. The first row shows in a coronal plane the resulting dipoles of the ECD, which revealed a source in the right temporal pole. The second row depicts a cortical rendering of the patient’s MRI, seen in the right hemisphere. The result of the DSM also reveals a source in the right temporal pole. (**B**) Interictal LTM, ictal and interictal HD sources from a patient with extratemporal lobe epilepsy who previously had a resection in the right parietal lobe. The results of the ECD (first row, sagittal plane) and the DSM (second row, superior view on the cortical rendering) show sources around the previous resection cavity. (LTM: long-term monitoring, HD: high-density EEG, ECD: equivalent current dipole modelling, DSM: distributed source modelling).

**Figure 4 diagnostics-12-02303-f004:**
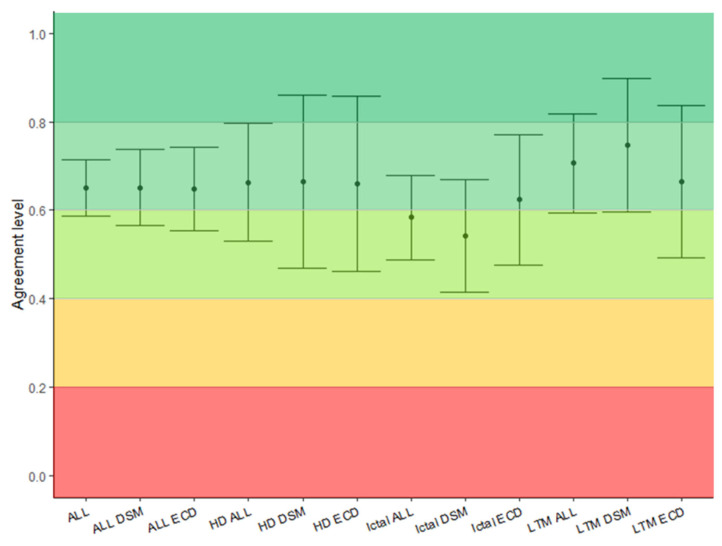
The interval plot shows the agreement level of each ESI modality. On the y axis, AC values and 95% confidence intervals are reported. On the x axis, ESI modalities are reported. Analysers obtained substantial agreement in all ESIs modalities. Agreement level: <0: poor, 0.01–0.2: slight, 0.2–0.4: good, 0.4–0.6: moderate, 0.6–0.8: substantial, 0.8–1: almost perfect. The colour map underlines the level of agreement from red (poor) to dark green (almost perfect). (AC: agreement coefficient, ECD: equivalent current dipole, DSM: distributed source model, HD: high density, LTM: long term monitoring).

**Table 1 diagnostics-12-02303-t001:** Gwet AC1 values and 95% confidence intervals. Analysers showed substantial agreement among ESI modalities. Ictal analysis showed slightly inferior agreement, but without significant difference. Rows: analysed signal, columns: inverse solutions. Results are reported as AC1 value [confidence interval].

	Overall	ECD	DSM
**Overall**	0.65 [0.59–0.71]	0.65 [0.56–0.74]	0.65 [0.57–0.74]
**Interictal HD**	0.66 [0.53–0.80]	0.66 [0.46–0.86]	0.66 [0.47–0.86]
**Interictal LTM**	0.71 [0.60–0.82]	0.67 [0.49–0.89]	0.75 [0.60–0.90]
**Ictal**	0.58 [0.50–0.68]	0.62 [0.48–0.77]	0.54 [0.42–0.67]

AC: agreement coefficient, ECD: equivalent current dipole, DSM: distributed source model, HD: high density, LTM: long term monitoring. Agreement level: 0–0.2: fair, 0.2–0.4: good, 0.4–0.6: moderate, 0.6–0.8: substantial, 0.8–1: almost perfect.

## Data Availability

The data that support the findings of this study are available from the corresponding author upon reasonable request.

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
