# Peer review of "Electric Source Imaging in Presurgical Evaluation of Epilepsy: An Inter-Analyser Agreement Study"

_diagnostics, 2022, doi:10.3390/diagnostics12102303_

Round 1

Reviewer 1 Report

The manuscript describes an inter-expert comparison of presurgical evaluation of epilepsy patients. Although a quite big dataset and expert group, all rely on the same software and pipeline, so we do not know if the method is appropriate. It is not surprising that there is not as big difference between experts from the same center, trained from similar schools, and when they used the same pipeline. The value of the study will be much bigger if there is at least included another practice is taken into account: software or analyzers. Therefore I would not recommend the paper for publication.

Author Response

Point-by-point answers to reviewer’s comments.

Dear Editor

We thank the reviewers for their helpful comments.We have addressed all comments and suggestions. We clarified and elaborated on our research question, added general information on the management of epilepsy in the introduction, elaborated on the analysers’ background and discussed properly the limitations of our study. Our detailed response to the comments of the reviewer is below.

Yours sincerely

Sándor Beniczky MD, PhD, FEAN

Legend:

Plain font indicates the reviewer comments

Italic font indicates the reply to the reviewer.

The new sentences in the revised manuscript are written in red.

We have addressed the question suggested here by the reviewer in a previous publication from our group (Duez et al, Neurology 2019). This was specified in the Discussion section. In our previous paper we showed that only moderate agreement could be reached using different software and analyzers. That result raised the question whether standardization of the methodology, using the same pipeline and a software approved for medical use, would improve the agreement? Is it possible to teach ESI to trainees / fellows, so that they achieve the same results? We believe that these questions addressed in our present study, are important, with relevance for clinical implementation. Reviewers 2 and 3 agreed on this, and they appreciated the importance of our work.

We clarified our research question in the introduction section on page 1 and the abstract. We extensively described experts’ previous training in the methods section, page 3. We also added a “limitations” paragraph in the discussion section on page 10 to explain the use of a single software in this study.

Abstract, page 1

Six experts, of whom five had no previous experience in ESI, independently performed interictal and ictal ESI of 25 consecutive patients (17 temporal, eight extratemporal) who underwent presurgical evaluation. The overall agreement among experts for the ESI methods was substantial (AC1 = 0.65; 95% CI: 0.59-0.71), and there was no significant difference between the methods. Our results suggest that using a standardised analysis pipeline, newly trained experts reach similar ESI solutions, calling for more standardisation in this emerging clinical application in neuroimaging.

Introduction, page 1

Obviously, this method would be of little or no clinical use, if different experts reached different results when analysing the same recordings, using the same pipeline and software. Nonetheless, this important aspect has not been investigated yet. The aim of this study is therefore to investigate the degree of inter-analyser agreement of ESI performed, using a standardized pipeline and a software certified for medical use, by young experts without previous experience in performing ESI.

Methods, page 3

Six experts (SB, TC, AC, LH, EC, PM) independently applied the same analysis pipeline, while selection of IEDs, selection of ictal waves and of the signal time-window were analyser dependent, as these aspects are tailored to the individual patient. SB had previous experience (18 years) in ESI, while the other analysers were introduced to the methodology at the start of this study. Two of them (TC, EC) were senior experts with more than five years of experience in EEG reading. Three analysers were junior experts with less than five years of experience in EEG reading (AC, PM, LH). All experts had their EEG training in different centers in Europe.

Reviewer 2 Report

In the present manuscript authors investigated an important issue in daily clinical practice in Epilepsy centers delaying with refractory epilepsies and identification of epileptogenic focus in the brain.  Epileptic source imaging Interanalyser agreement was high enough to support the reproducibility of this method. 

Overal, manuscript is of the interest for the epileptologists and is of scientific merit. Nicely illustrated. 

Minor comments to improve the study presentation: 

- Introduction would benefit if you add several sentences on the refractory epilepsy and its management, as well on basic facts of epilepsy diagnosis in order to make these facts clear to wider readers. 

- Although experts included in the study are described in some details, it would be beneficial to provide data on their educational background and years of experience in EEG reading and ESI method application (younger vs. seniors). It could be done in the text only without additional graphical elements. 

-Discussion should include comments on limitations of the study, with focus on the number of observers and cases (it could be higher).

Author Response

Point-by-point answers to reviewer’s comments.

Dear Editor

We thank the reviewers for their helpful comments.We have addressed all comments and suggestions. We clarified and elaborated on our research question, added general information on the management of epilepsy in the introduction, elaborated on the analysers’ background and discussed properly the limitations of our study. Our detailed response to the comments of the reviewer is below.

Yours sincerely

Sándor Beniczky MD, PhD, FEAN

Legend:

Plain font indicates the reviewer comments

Italic font indicates the reply to the reviewer.

The new sentences in the revised manuscript are written in red.

REVIEWER 2

In the present manuscript authors investigated an important issue in daily clinical practice in Epilepsy centers delaying with refractory epilepsies and identification of epileptogenic focus in the brain.  Epileptic source imaging Interanalyser agreement was high enough to support the reproducibility of this method.

Overal, manuscript is of the interest for the epileptologists and is of scientific merit. Nicely illustrated.

Minor comments to improve the study presentation:

- Introduction would benefit if you add several sentences on the refractory epilepsy and its management, as well on basic facts of epilepsy diagnosis in order to make these facts clear to wider readers.

Reply:

We agree with this comment. We added the paragraph below as the first paragraph to the introduction on page 1 of the manuscript.

Refractory epilepsy has been defined by the International League Against Epilepsy as the persistence of seizures even after adequate trials with two or more appropriate and tolerated anti seizure drugs [1]. The occurrence of seizures lower the quality and quantity of life in affected patients and increase the risk of Sudden Unexpected Death in Epilepsy [2]. For patients with refractory focal epilepsy, resective surgery is an evidence-based threatment option to obtain seizure freedom [3,4]. Epilepsy surgery is only successful if the so-called epileptogenic zone can be reliably identified and safely resected. In order to estimate the location and extent of the epileptogenic zone, patients undergo a multimodal presurgical workup, including long-term video-EEG monitoring, structural (MRI) and functional (FDG-PET, ictal/interictal SPECT, fMRI, DTI) neuroimaging, neuropsychological testing and occasionally invasive EEG monitoring [3,5]. A relatively new and underused technique during the presurgical evaluation is electric source imaging (ESI).

- Although experts included in the study are described in some details, it would be beneficial to provide data on their educational background and years of experience in EEG reading and ESI method application (younger vs. seniors). It could be done in the text only without additional graphical elements.

Reply:

This is a good point, and we thank the reviewer for the suggestion. The only expert who has previous experience with ESI is autor SB. The other experts were introduced to the methodology at the start of this project. SB is a board-certified neurologist, clinical neurophysiologist and epileptologist with 20 years of experience in EEG reading and 18 years of experience with ESI, working in Denmark. TC is a pediatric neurologist with more than 10 years of experience at an epilepsy center in Germany, where he is responsible for the presurgical evaluation of adult and pediatric patients with refractory epilepsy. EC has a PhD in Biomedical Sciences and has 6 years of experience in Belgium in functional multimodal neuroimaging during the presurgical evaluation of adult and pediatric patients, including EEG reading. AC is a senior resident in child’s neuropsychiatry in Italy and has 4 years of experience in EEG reading. LH is a MD, last year PhD researcher in the field of epilepsy (seizure detection, prediction tools, EEG biomarkers) and first year neurology resident, with EEG training in Denmark and Hungary. PM is a senior resident in neurology in Italy and has 4 years of experience in EEG reading.

We elaborated on the background and level of experience in the methods section on page 3 as follows:

Six experts (SB, TC, AC, LH, EC, PM) independently applied the same analysis pipeline, while selection of IEDs, selection of ictal waves and of the signal time-window were analyser dependent, as these aspects are tailored to the individual patient. SB had previous experience (18 years) in ESI, while the other analysers were introduced to the methodology at the start of this study. Two of them (TC, EC) were senior experts with more than five years of experience in EEG reading. Three analysers were junior experts with less than five years of experience in EEG reading (AC, PM, LH). All experts had their EEG training in different centers in Europe.

-Discussion should include comments on limitations of the study, with focus on the number of observers and cases (it could be higher).

Reply:

We added the paragraph below to the discussion on page 10.

This study has some limitations. First, the sample size is relatively small, therefore the results should be confirmed by larger, multicentric studies.  Second, five of the experts had their ESI training in the same center. This was however part of the aim of our study, as we aimed at verifying the reliability of the methodology when applying the same pipeline while allowing subjective decisions that relied on the level of EEG experience of the experts (which was gained at different European centers). Third, the same software was used to perform the analysis, but the reliability of the use of different softwares in performing ESI has already been explored in a previous study by our group [10].

Reviewer 3 Report

Regarding the review of MS entitled "Electric Source Imaging in presurgical evaluation of epilepsy: an inter-analyser agreement study" I have found that MS is novel and sound. I have recommended to publication in its present form.

Author Response

Point-by-point answers to reviewer’s comments.

Dear Editor

We thank the reviewers for their helpful comments.We have addressed all comments and suggestions. We clarified and elaborated on our research question, added general information on the management of epilepsy in the introduction, elaborated on the analysers’ background and discussed properly the limitations of our study. Our detailed response to the comments of the reviewer is below.

Yours sincerely

Sándor Beniczky MD, PhD, FEAN

Legend:

Plain font indicates the reviewer comments

Italic font indicates the reply to the reviewer.

The new sentences in the revised manuscript are written in red.

REVIEWER 3

Regarding the review of MS entitled "Electric Source Imaging in presurgical evaluation of epilepsy: an inter-analyser agreement study" I have found that MS is novel and sound. I have recommended to publication in its present form.

Reply:

We thank the reviewer for appreciating our work.

Round 2

Reviewer 1 Report

Thank you for explaining the planned methodology and the importance of the work. After taking into account that all experts are not from the some school, had their EEG training in different centers in Europe, and even newly trained expert, it makes sense to publish such a study.